# Gut Dysbiosis Shaped by Cocoa Butter-Based Sucrose-Free HFD Leads to Steatohepatitis, and Insulin Resistance in Mice

**DOI:** 10.3390/nu16121929

**Published:** 2024-06-18

**Authors:** Shihab Kochumon, Md. Zubbair Malik, Sardar Sindhu, Hossein Arefanian, Texy Jacob, Fatemah Bahman, Rasheeba Nizam, Amal Hasan, Reeby Thomas, Fatema Al-Rashed, Steve Shenouda, Ajit Wilson, Shaima Albeloushi, Nourah Almansour, Ghadeer Alhamar, Ashraf Al Madhoun, Fawaz Alzaid, Thangavel Alphonse Thanaraj, Heikki A. Koistinen, Jaakko Tuomilehto, Fahd Al-Mulla, Rasheed Ahmad

**Affiliations:** 1Dasman Diabetes Institute, Dasman 15462, Kuwait; shihab.kochumon@dasmaninstitute.org (S.K.); mohammad.malik@dasmaninstitute.org (M.Z.M.); sardar.sindhu@dasmaninstitute.org (S.S.); hossein.arefanian@dasmaninstitute.org (H.A.); texy.jacob@dasmaninstitute.org (T.J.); fatemah.bahman@dasmaninstitute.org (F.B.); rasheeba.iqbal@dasmaninstitute.org (R.N.); amal.hasan@dasmaninstitute.org (A.H.); reeby.thomas@dasmaninstitute.org (R.T.); fatema.alrashed@dasmaninstitute.org (F.A.-R.); steve.shenouda@dasmaninstitute.org (S.S.); ajit.wilson@dasmaninstitute.org (A.W.); shaima.albeloushi@dasmaninstitute.org (S.A.); nourah.almansour@dasmaninstitute.org (N.A.); ghadeer.alhamar@dasmaninstitute.org (G.A.); ashraf.madhoun@dasmaninstitute.org (A.A.M.); fawaz.alzaid@inserm.fr (F.A.); alphonse.thangavel@dasmaninstitute.org (T.A.T.); fahd.almulla@dasmaninstitute.org (F.A.-M.); 2Université Paris Cité, INSERM UMR-S1151, CNRS UMR-S8253, Institut Necker Enfants Malades, F-75015 Paris, France; 3Department of Medicine, University of Helsinki and Helsinki University Hospital, 00029 Helsinki, Finland; heikki.koistinen@helsinki.fi; 4Department of Public Health and Welfare, Finnish Institute for Health and Welfare, P.O. Box 30, 00271 Helsinki, Finland; jaakko.tuomilehto@thl.fi; 5Minerva Foundation Institute for Medical Research, 00290 Helsinki, Finland; 6Department of Public Health, University of Helsinki, 00014 Helsinki, Finland

**Keywords:** cocoa butter, fish oil, high-fat diet, gut dysbiosis, steatohepatitis, insulin resistance

## Abstract

Background: High-fat diets cause gut dysbiosis and promote triglyceride accumulation, obesity, gut permeability changes, inflammation, and insulin resistance. Both cocoa butter and fish oil are considered to be a part of healthy diets. However, their differential effects on gut microbiome perturbations in mice fed high concentrations of these fats, in the absence of sucrose, remains to be elucidated. The aim of the study was to test whether the sucrose-free cocoa butter-based high-fat diet (C-HFD) feeding in mice leads to gut dysbiosis that associates with a pathologic phenotype marked by hepatic steatosis, low-grade inflammation, perturbed glucose homeostasis, and insulin resistance, compared with control mice fed the fish oil based high-fat diet (F-HFD). Results: C57BL/6 mice (5–6 mice/group) were fed two types of high fat diets (C-HFD and F-HFD) for 24 weeks. No significant difference was found in the liver weight or total body weight between the two groups. The 16S rRNA sequencing of gut bacterial samples displayed gut dysbiosis in C-HFD group, with differentially-altered microbial diversity or relative abundances. *Bacteroidetes*, *Firmicutes*, and *Proteobacteria* were highly abundant in C-HFD group, while the *Verrucomicrobia*, *Saccharibacteria* (TM7), *Actinobacteria*, and *Tenericutes* were more abundant in F-HFD group. Other taxa in C-HFD group included the *Bacteroides*, *Odoribacter*, *Sutterella*, *Firmicutes bacterium* (AF12), *Anaeroplasma*, *Roseburia*, and *Parabacteroides distasonis.* An increased Firmicutes/Bacteroidetes (F/B) ratio in C-HFD group, compared with F-HFD group, indicated the gut dysbiosis. These gut bacterial changes in C-HFD group had predicted associations with fatty liver disease and with lipogenic, inflammatory, glucose metabolic, and insulin signaling pathways. Consistent with its microbiome shift, the C-HFD group showed hepatic inflammation and steatosis, high fasting blood glucose, insulin resistance, increased hepatic de novo lipogenesis (Acetyl CoA carboxylases 1 (*Acaca*), Fatty acid synthase (*Fasn*), Stearoyl-CoA desaturase-1 (*Scd1*), Elongation of long-chain fatty acids family member 6 (*Elovl6*), Peroxisome proliferator-activated receptor-gamma (*Pparg*) and cholesterol synthesis (β-(hydroxy β-methylglutaryl-CoA reductase (*Hmgcr*). Non-significant differences were observed regarding fatty acid uptake (Cluster of differentiation 36 (*CD36*), Fatty acid binding protein-1 (*Fabp1*) and efflux (ATP-binding cassette G1 (*Abcg1*), Microsomal TG transfer protein (*Mttp*) in C-HFD group, compared with F-HFD group. The C-HFD group also displayed increased gene expression of inflammatory markers including Tumor necrosis factor alpha (*Tnfa*), C-C motif chemokine ligand 2 (*Ccl2*), and Interleukin-12 (*Il12*), as well as a tendency for liver fibrosis. Conclusion: These findings suggest that the sucrose-free C-HFD feeding in mice induces gut dysbiosis which associates with liver inflammation, steatosis, glucose intolerance and insulin resistance.

## 1. Introduction

Diet and gut flora have a reciprocal relationship, which occurs when the host’s dietary components influence the microbiota [1]. Recent evidence shows that dietary fats can affect how gut microorganisms function and affect the body’s ability to break down certain food components. Moreover, microbiota synthesize certain amino acids and vitamins, which sustain the host’s health [2]. The gut dysbiosis from dietary or environmental factors can result in the overgrowth of pathogenic bacteria that cause chronic inflammation, contributing significantly to the pathology of chronic metabolic disorders such as obesity [3], type 2 diabetes [4,5], liver diseases [6], and intestinal diseases [7,8].

The human microbiome is comprised of two major phyla, including *Bacteroidetes*, and *Firmicutes*. The *Bacteroidetes* phylum includes four major classes, namely *Bacteroidia*, *Flavobacteria*, *Sphingobacteria*, and *Cytophagia*, all of which ferment indigestible carbohydrates [9]. The *Firmicutes* are classified into three major classes including *Clostridium*, *Negativicute*, and *Bacillus*, which generate short-chain fatty acids [10]. Several studies have shown that the altered *Firmicutes*/*Bacteroidetes* (F/B) ratio is a biomarker for dysbiosis and suggested that a higher F/B ratio might relate to obesity in animals and humans [11,12]. However, others reported that the F/B ratio had no association with gut microbial alteration patterns and with microbiome functional shift [13]. As a result, interest has increased in understanding the effects of dietary fats on gut microbiome. Dietary fat provides energy, and there are many major food sources of dietary fats. Lard and palm oil are high in saturated fatty acids which are considered bad for health [14]. Fish oil and olive oil are high in unsaturated fatty acids which are considered good for health. Excessive consumption of saturated fats substantially impacts the intestinal microbiota and body weight. Conversely, a higher consumption of unsaturated fatty acids is regarded as beneficial for health [15]. Cocoa butter is widely in the form of chocolate and the metabolic and physiological effects of this fat remain unclear. Cocoa butter and fish oil are the dietary fats that most likely have different health effects. Cocoa butter is extracted from the cocoa bean and it mainly contains saturated fat (24–30% palmitic acid and 24–37% stearic acid), 30–38% monounsaturated fat (oleic acid), and 1% polyunsaturated fat (linoleic acid), in addition to low levels of lauric acid and myristic acid [16]. Whereas, fish oil is rich in omega-3 fatty acids which are known to have health benefits owing to their anti-inflammatory and anti-cancer effects [17]. It was shown that cocoa butter-based high-fat feeding in rats for three days did not affect plasma total triglyceride (TG) levels or TG-rich very low-density lipoprotein (VLDL) particles or hepatic insulin sensitivity, but the changes in hepatic gene expression led to increased lipid synthesis, lipotoxicity, inflammation, and insulin resistance [18].

Differential effects of fat sources have been shown to influence health via the gut-liver axis. The dietary and gut microbiota-induced changes in the liver have been shown to affect both the biosynthesis and degradation of hepatic lipids [19,20,21]. In addition, dietary carbohydrates are also known to modulate gut microbiota and influence metabolic health [22]. The aim of the study was to compare the effects of two different dietary fats in absence of sucrose, i.e., cocoa butter and fish oil, on the gut microbiota shift or dysbiosis in mice that were fed these two types of high fat diets (HFDs) for 24 weeks. We also examined the clinical outcomes in relation to hepatic steatosis, low-grade inflammation, glucose tolerance, and peripheral insulin sensitivity. Here, we report that the cocoa butter-based high fat diet (C-HFD) feeding in mice induces a shift in the gut microbiome which associates with liver inflammation, steatosis, tendency for hepatic fibrosis, loss of glucose tolerance, and insulin resistance whereas the control mice fed with fish oil-based high fat diet (F-HFD) did not develop gut dysbiosis and these mice remained metabolically healthy.

## 2. Materials and Methods

### 2.1. Animals

Animal experimental procedures were performed following the National Institutes of Health guidelines for the care and use of laboratory animals and were reviewed and approved by the Institutional Animal Care and Ethics Committee (Approval No. RA AM 2016-007). C57BL/6 mice were purchased from the Jackson Laboratory and the animals were bred in the institutional animal facility and fed ad libitum on standard chow diet. Mice were housed in a temperature-controlled room (23 °C) and maintained on a 12-h dark/12-h light cycle. All experiments were performed using 8–10 weeks old male mice. The mice were randomly divided into 2 groups, 5–6 mice per group per cage, and body weight was recorded. Two types of sucrose-free high fat diets (45% kcal fat) were used: i.e., cocoa butter based experimental diet (D18060406, Research Diets Inc., New Brunswick, NJ, USA) and fish oil based diet (D18060407, Research Diets Inc., New Brunswick, NJ, USA). Body weight and food intake of mice were recorded on a weekly basis. Fish oil based HFD cohort is being used as a control for comparison for various studies conducted simultaneously with the various HFDs based on different fat sources. At 21–22 weeks of dietary intervention, oral glucose tolerance test (OGTT) and insulin tolerance test (ITT) were performed. The mice were sacrificed at the end of dietary intervention (24 weeks) and all tissues and organs were collected and flash frozen in liquid nitrogen. Blood was collected for isolation of plasma. Plasma samples were stored at −80 °C until use. Liver tissue was fixed in 10% neutral buffered formalin and preserved by paraffin embedding. The fecal content was also aseptically collected from each mouse and stored at −80 °C until use.

### 2.2. Oral Glucose Tolerance Test (OGTT)

To perform the glucose tolerance test at 21 weeks feeding, mice were fasted for 12 h. Glucose (1 g/kg body weight) was administered orally to each mouse and blood glucose levels at different time points were measured at 0, 10, 20, 30, 60, 90, and 120 min using a portable glucometer.

### 2.3. Insulin Tolerance Test (ITT)

To determine insulin tolerance at 22 weeks feeding, mice were fasted for 4 h and insulin (0.8 U/kg body weight) was administered intraperitonially. Blood glucose levels were determined at 0, 15, 30, 45, 60, 75, 90, 105, and 120 min using a portable glucometer.

### 2.4. Plasma Measurements

Plasma metabolic hormone levels including insulin, c-peptide, glucagon, amylin, leptin, PYY, PP, ghrelin, GLP-1, GIP, and resistin were measured using a MILLIPLEX kit (MILLIPLEX kit MAP mouse Metabolic Magnetic Bead Panel, Millipore, Burlington, MA, USA) in accordance with the manufacturer’s instructions.

### 2.5. Histological Analysis

Histology analysis was performed using standard protocols. Detailed methods of Histological Analysis were described in the Appendix A.

Hematoxylin-Eosin (H&E) staining and for Oil Red O staining for fat content following standard protocols [23,24,25,26,27] as described in the Appendix A.

### 2.6. Gene Expression Analysis by qRT-PCR

Total RNA was extracted from liver tissue using RNeasy Mini kit (Qiagen, Hilden, Germany). 1 μg of total RNA were reverse transcribed into cDNA using the High Capacity cDNA Reverse Transcription Kit (Applied Biosystems, Foster City, CA, USA). Quantitative real-time PCR was performed in a QuantStudio™ 5 Real-Time PCR System using TaqMan master mix reagents and gene specific TaqMan assays (Applied Biosystems, Foster City, CA, USA). Each reaction was performed in triplicate using standard reaction conditions. Target gene cycle threshold (Ct) values were normalized against GAPDH Ct values and gene expression level relative to control was calculated using 2^−ΔΔCT^ method. The gene specific primers are listed in Appendix A.

### 2.7. Microbiome Sequencing

DNA extraction from mouse fecal samples was carried out using Qiamp DNA fast stool mini kit Qiagen, Germany and quantified using a Qubit fluorometer (Thermo Fisher Scientific, Waltham, MA, USA), following the manufacturer’s instructions. A total of 5 ng/μL of microbial DNA was amplified using gene-specific primers that target the bacterial 16S rRNA V3 and V4 regions. The full-length primers with overhang adapter sequences used are as follows: 16S Amplicon PCR Forward Primer: 5′ TCGTCGGCAGCGTCAGATGTGTATAAGAGACAGCC TACGGGNGGCWGCAG; 16S Amplicon PCR Reverse Primer: 5′ GTCTCGTGGGCTCGGAGATGTGTAT AAGAGACAGGACTACHVGGGTATCT AATCC. PCR reaction was carried out using KAPA HiFi HotStart Ready Mix PCR mix kit (Roche Diagnostics, Indianapolis, IN, USA), following the manufacture’s protocol. The resulting PCR product was confirmed on a Bioanalyzer using high sensitivity DNA chip (Agilent Technologies, Inc., Santa Clara, CA, USA), purified using AMPure XP beads (Agilent Technologies, Inc., Santa Clara, CA, USA) and bound to dual indices using the Nextera XT Index Kit (Illumina Inc., San Diego, CA, USA), following the manufacturers’ recommendations. Purified and normalized libraries were multiplexed up to 24 samples and paired-end sequencing was carried out using MiSeq platform (Illumina Inc., San Diego, CA, USA).

### 2.8. 16S rRNA Microbiome Analysis and Bioinformatics Statistics

The sequence data obtained were analyzed using QIIME 2 (Quantitative Insights Into Microbial Ecology) version 2022.8 [28,29], https://www.nature.com/articles/s41598-019-44369-y (accessed on 1 April 2023) ref-CR26 and MicrobiomeAanalyst package [30]. Using QIIME 2, the forward and reverse reads of the same sample were first joined. The paired reads were demultiplexed and quality filtered at Q-score of 25. High-quality amplicon sequence variants (ASVs) were obtained by using the Divisive Amplicon Denoising Algorithm-2 (DADA2) algorithm [31]. The taxonomy profile analysis was performed against the Greengenes reference database [32] and transformed into relative abundance at the phylum, class, order, family, genus, and species levels. After this, using Greengenes reference database, open-reference operational taxonomic units (OTUs) were picked out from the non-chimeric sequences at 97% similarity. The most abundant read from each OTU was selected as the representative read for that OTU. The taxonomy associated with Greengenes database to which OTUs matched was assigned to OTUs. To test two-group differences in the percentage of analyzable read numbers between HFF and HFS, we calculated *p*-values using the Wilcoxon signed-rank test. The OTU table was filtered to remove the OTUs containing <10 counts in all samples and were transformed into relative abundances at the phylum, class, order, family, genus, and species levels.

### 2.9. Biodiversity Analysis

The intra-sample (alpha) and the inter-sample (beta) diversity analyses were performed as described elsewhere [33,34,35]. The alpha diversity was calculated using three different measures: Observed and Fisher statistics and phylogenetic diversity [36], performed as the references cited for each. The bacterial genus/spp. data were provided. Beta diversity was calculated using Bray-Curtis distances matrices, generating the two-dimensional Principal Coordinate Analysis (PCoA) plots. Weighted and un-weighted UniFrac distance matrices were used to derive the beta diversity of the samples [33] which data could be accessed. Non-parametric Mann-Whitney/Kruskal-Wallace tests were used to determine the statistical significance of α-diversity measures and permutational MANOVA was used to determine the differences in β-diversity.

### 2.10. Identification of Biomarker Microbiome

Linear discriminant analysis (LDA) effect size (LEfSe) analysis was performed to identify the bacteria for which the relative abundance was significantly increased or decreased in each phenotypic category. LEfSe algorithm using the Benjamini-Hochberg false discovery rate (FDR) adjusted *p*-value cutoff of 0.05 and the logarithmic LDA score cutoff of 2.0. The LEfSe bar plots were created using the MicrobiomeAnalyst package [30]. In all analyses, *p*-values were corrected for Benjamini-Hochberg FDR. Random Forest analysis was applied to the 16S rRNA data using machine learning methods developed by using the MicrobiomeAnalyst package to identify the most significant microbial features. Features with a minimum prevalence of 10% across samples were included, and those with >0.005 accuracy were considered significant. Data were further transformed to cantered log ratio (CLR) before applying the Random Forest classification algorithm.

### 2.11. Microbial Metagenomic Functional Predictions

Phylogenetic investigation of communities by reconstructing unobserved states-2 (PICRUSt2) software to predict the microbial content from each gut microbe sample and functionally annotate the data [37]. The results were further analyzed using the Microbiomeanalyst package [30]. To investigate the metabolic network of the predicted organisms, we used the MetaCyc database [38], which contains data regarding chemical compounds, reactions, enzymes, and metabolic pathways that have been experimentally validated. The correlation between microbial community and their metabolites was performed using a Microbiomeanalyst package [30].

### 2.12. Prediction of Biomarker Microbiome-Epigenome Interactions

The human microbiome affects the host gene expression as the microbiota-secreted proteins, microbiota-derived components, and microbiota-derived metabolites may regulate the host’s physiology by modifying its gene expression. Therefore, biomarker microbiome-epigenome interactions were predicted by using the Microbiomeanalyst taxon set enrichment analysis [30], Human Microbe-Disease Association Database (HMDAD) [38], and the human microbiome affect the host epigenome (MIAOME) database [39,40].

## 3. Results

### 3.1. C-HFD and F-HFD Have Differential Effects on the Gut Microbiome in Mice

High-fat diets containing sucrose have been associated with overweight and obesity, except for fish oil based high fat diets [41]. Since dietary sucrose content is known to affect both adiposity [42] and gut microbiota [43,44], we wanted to compare the effects of two types of sucrose-free HFDs, i.e., C-HFD and F-HFD on the gut microbiome of mice that were fed with each diet individually for a period of 24 weeks. Notably, our data show that the mice fed with sucrose-free C-HFD showed no significant difference in the liver and total body weights compared to those fed with F-HFD mice (Appendix A).

To identify the gut microbiota changes in C-HFD and F-HFD fed mice, fecal DNA samples were sequenced for 16S rRNA V3/V4 genes using high-throughput Illumina MiniSeq, which generated a total of 4819433 reads, with an average of 481,943 reads per sample. For quality filtering, 20% of reads across all samples were removed and data were tested by quality control procedures. The rarefaction curve data show the sequencing depth as a measure of microbial diversity found in the samples analyzed (Figure 1A). Alpha diversity matrices including the observed number of operational taxonomic units (OTUs), and Fisher’s alpha diversity index were significantly different, each, while comparing C-HFD and F-HFD mice groups (Figure 1B). Principal component analysis (PCA) showed clear separation of the microbial communities analyzing by diet, as measured by using unweighted UniFrac distances (PERMANOVA, *p* < 0.05) (Figure 1C).

To determine the microbial distribution in C-HFD and F-HFD groups, first we investigated the differential taxonomic abundance of microbiota at different levels. Plots of relative bacterial abundances (%) at the phylum, class, order, family, genus, and species levels were different between C-HFD and F-HFD mice groups (Figure 1D–H). We found that relative abundances of 8 bacteria at the phylum level, 33 bacteria at the genus level, and 12 bacteria at the species level differed significantly (Mann-Whitney U test, *p* < 0.05) between C-HFD and F-HFD fed mice. Compared with F-HFD group, gut microbiota from C-HFD mice had an increased abundance of 3 phyla including Firmicutes, Bacteroidetes, and Proteobacteria, while the F-HFD mice showed relative abundances of 4 phyla including *Verrucomicrobia*, *TM7*, *Actinobacteria*, and *Tenericutes* (Figure 1D). Next, at the genus level, C-HFD mice showed a higher abundance of *Bacteroides*, *Odoribacter*, *Sutterella*, *AF12*, *Anaeroplasma*, and *Roseburia* but a lower abundance of *Coprococcus*, *Lactobacillus*, *Anaerotruncus*, *Turicibacter*, *Clostridium*, *Sporosarcina*, and *Rikenella*, as compared with mice F-HFD mice (Figure 1G). We also determined the relative abundances at the species level and found that C-HFD mice had higher abundances of *Bacteroides acidifaciens*, *Parabacteroides distasonis*, and *Bacteroides uniformis* but lower abundances of *Mucispirillum schaedleri* and *Akkermansia muciniphila* (Figure 1H–J) We next investigated the differences in gut microbiota composition between two dietary groups. The compositional method was used to search for distance patterns to identify the correlations of taxa with C-HFD and F-HFD feeding. This correlation was calculated using comparison of relative abundance through the linear discriminant analysis (LDA) effect size (LEfSe). LDA values of >2 with *p* < 0.05 were considered significantly enriched. A total of 20 taxa were found to have changed significantly, among which, 9 taxa were of higher abundance in C-HFD group compared with F-HFD group, which included *Bacteroides acidifaciens*, *Odoribacter*, *Sutterella*, *AF12*, *RF32*, *Anaeroplasma*, *Parabacteroides distasonis*, *Bacteroides uniformis*, and *Roseburia*, whereas 12 taxa had lower abundance which included *4C0d_2 (YS2)*, *Coprococcus*, *Lactobacillus*, *Anaerotruncus*, *Turicibacter*, *Clostridium*, *schaedleri*, *Sporosarcina*, *F16*, *Rikenella*, *C21_c20*, and *S24_7* in C-HFD fed mice (Figure 1K,L). Based on fold change (Log_2_ counts) comparisons of the differential abundance of key gut microbiota, *Bacteroides uniformis*, *Parabacteroides distasonis*, *RF32*, *AF12*, *Bacteroides acidifaciens*, and *Anaeroplasma* were of higher abundance, while *Mucispirillum schaedleri*, *C21-c20*, *Rikenella*, *Anaerotruncus*, and *Turicibacter* were of lower abundance in C-HFD group as compared with F-HFD group (Figure 1M).

To understand how the gut microbiota were predicted for disease association, we explored the associations between gut microbiome and metabolic diseases. These predicted associations between diseases and the biomarker gut microbiota were selected based on previous literature findings. In this regard, biomarker gut microbes that were differentially altered between C-HFD and F-HFD groups were investigated for their networks of association with various diseases. Among highly abundant gut microbial taxa in C-HFD fed mice, *Sutterella*, *Anaeroplasma*, *Bacteroides acidifaciens*, *Roseburia*, *Odoribacter*, and *AF12* had predicted associations with metabolic disorders including non-alcoholic fatty liver disease, lipid metabolism disorders, liver cirrhosis, insulin resistance, fatty liver, metabolic syndrome, and obesity (Figure 1N).

### 3.2. Biomarker Gut Microbiota Associate with Host Genes Involved in Immune-Metabolic Regulation

We sought to determine the associations between biomarker gut microbiota and the host genes linked to key immune-metabolic pathways. To this effect, we identified the host genes using MicrobiomeAnalyst taxon set enrichment analysis and MIAOME database [40]. To characterize the potential significance of the host gene-biomarker gut microbiota associations in mice fed C-HFD and F-HFD diets, we further analyzed the gut microbiota-host gene associations in perspective of their linked pathways (Figure 2A–C). The associations between differentially expressed biomarker gut microbiota and the predicted host pathways genes are shown (Figure 2A). The functional enrichment analysis of the predicted host genes revealed that the host genes were linked to critical pathways associated with immune-metabolic disorders. These significant pathways were associated with the inflammatory and immune responses, cytokine production including expression/regulation of interleukin (IL)-1β, IL-6, IL-8, IL-10, and IL-12 signaling, insulin resistance, cellular responses to lipopolysaccharide (LPS), steroids biosynthesis, apoptosis, and the cascades involving nuclear factor kappa B (NF-κB), NLR family pyrin domain containing 3 (NLRP3), extracellular signal-regulated kinase 1/2 (ERK1/2), c-Jun *N*-terminal kinase (JNK), mitogen-activated protein kinases (MAPKs), and MyD88-TLR mediated signaling (Figure 2B). The pathway enrichment analysis showed that the microbial taxa including Bacteroides uniformis, Clostridium, Bacteroides acidifaciens, Corprococcus, Mucispirillum schaedleri, Rikenella, and Lactobacillus were associated with host genes such as *CD44*, *CB4*, nucleotide-binding oligomerization domain 1/2 (*NOD1/2*), *TLR4/5/9*, *NLRP3*, melanocortin 4 receptor (*MC4R*), and sirtuin-1 (*SIRT1*) in the context of inflammatory responses and non-alcoholic fatty liver disease (NAFLD) pathways (Figure 2C). Similarly, these microbial taxa (except *Bacteroides uniformis*) were also associated with the same host genes (excluding *CD44* and including *CYP27B1*) regarding cellular response to LPS and insulin resistance (Figure 2D).

The functional abundance of necessary bacterial enzymes was predicted using phylogenetic investigation of communities by reconstructing unobserved states (PICRUSt2) analysis. The metabolic enzymes involved in the gut were different in C-HFD and F-HFD-fed mice and predicted potential functional differences in terms of bacterial enzyme production and diet-induced changes in the gut microbiomes. Using PICRUSt2-based analysis, we determined which gut bacterial taxa were associated with significant changes in metabolic enzymes. To understand the relationships between gut microbiota and the bacterial enzymes or metabolites, Pearson’s correlation analysis of the differential gut microbiota between C-HFD and F-HFD groups and their metabolites was performed by using MicrobiomeAnalyst [30]. We found that the differentially-expressed higher abundance gut microbiota in C-HFD fed mice such as Parabacteroides distasonis, Bacteroides uniformis, Bacteroides acidifaciens, and Sutterella were strongly associated with Galactitol-1-phosphate 5-dehydrogenase, Carnitine 3-dehydrogenase, L-iditol 2-dehydrogenase, and 3-hydroxybutyryl-CoA dehydrogenase (Figure 2E). Gluconate 2-dehydrogenase, GDP-mannose 6-dehydrogenase, and L-idonate 5-dehydrogenase (NAD(P)(+)) were directly associated with 4C0d_2 (YS2), *Sporosarcina*, *Atopostipes*, and *Lactobacillus* (Figure 2). On the contrary, the lower abundance gut microbiota in C-HFD fed mice such as *Lactobacillus*, *Mucispirillum schaedleri*, *Anaerotruncus*, *Rikenella*, and *C21_C20* were strongly associated with L-idonate 5-dehydrogenase (NAD(P)(+)) sn-glycerol-1-phosphate dehydrogenase, Dihydrokaempferol 4-reductase, UDP-*N*-acetylglucosamine 6-dehyrogenase, Sorbitol-6-phosphate 2-dehydrogenase, Mannitol-1-phosphate 5-dehydrogenase, Alcohol dehydrogenase (NADP(+)), 3-dehydro-L-gulonate 2-dehydrogenase, and L-rhamnose 1-dehydrogenase (Figure 2E).

### 3.3. Mice Fed with C-HFD Display Liver Steatosis and Inflammation

We speculated that the pathological shift in gut microbiota observed in mice fed with C-HFD could lead to induce hepatic steatosis and liver inflammation in these mice. We next investigated the liver tissue phenotypes of mice fed these diets. As expected, the livers from C-HFD fed mice showed increased fat accumulation. No fat accumulation was observed in the livers of F-HFD fed mice (Figure 3A). Hepatic steatosis (microvesicular and macrovesicular) of mixed type was most obvious (Figure 3B,C). Hepatic lipid accumulation was confirmed by oil red O-staining of the liver sections, which revealed numerous lipid droplets that were not observed in the livers of mice fed with F-HFD (Figure 3D,E). Consistent with the development of hepatic steatosis, RT-qPCR analyses of the liver specimens from C-HFD mice indicated significantly upregulated expression of the genes involved in de novo lipogenesis (Acaca: acetyl CoA carboxylase; Fasn: fatty acid synthetase, Scd1: stearoyl Coa desaturase 1; Ppar-γ: Peroxisome proliferator-activated receptor gamma) and TG synthesis (Elovl6: Fatty Acyl-CoA Elongase), compared with F-HFD mice (Figure 3F). The expression of cholesterol synthesis associated genes (*Hmgcr*: 3-Hydroxy-3-Methylglutaryl-CoA Reductase, *Srebp1*: sterol regulatory element-binding protein 1) was also found to be increased in C-HFD mice (Figure 3F). However, no difference was observed in the mRNA expression of fat uptake genes *CD36* and *Fabp1* between the two groups of mice (Figure 3G). The low-density lipoprotein receptor (Ldlr) which plays a role in uptake of cholesterol was upregulated in the livers of C-HFD fed mice (Figure 3H). Fatty acid transport protein 1 (*Fatp1*) gene expression was downregulated in livers of mice fed with C-HFD (Figure 3H). No significant changes were observed in the expression of genes associated with fat efflux (*Abcg1*: ATP binding cassette G1, Mttp: microsomal triglyceride transfer protein) between two dietary groups (Figure 3I). Interestingly, mRNA expression of β-Oxidation genes carnitine palmitoyltransferase 1a (*Cpt1a*) and peroxisome proliferator activated receptor alpha (*Pparα*) were significantly elevated in C-HFD mice (Figure 3J).

Since the gut microbiome analyses suggested that microbial dysbiosis associated with inflammatory pathways, we assessed inflammation in the livers of the mice fed either with C-HFD or F-HFD. As expected, H&E staining of the liver specimens of C-HFD mice revealed presence of the lobular inflammation. However, no inflammation was seen in the livers of the mice fed with F-HFD (Figure 4A,B). Furthermore, C-HFD feeding resulted in the accumulation of F4/80-positive macrophages (Figure 4C,D), indicating active liver inflammation. Consistent with the infiltration of macrophages and lobular inflammation, genes related to liver inflammation (Tnf-α, Ccl2 and Il12b) were significantly upregulated in C-HFD mice compared with F-HFD mice (Figure 4E). However, no significant difference was observed in the expression of Il-1β and Il6 between the two dietary groups (Figure 4E). Trichrome blue-green staining of collagen fibers was performed to determine the relative extent of fibrosis in the C-HFD mice livers compared with F-HFD mice livers which indicated a trend in fibrosis in the livers of C-HFD mice, but it was not statistically significant (Figure 4F,G).

### 3.4. Impaired Glucose Tolerance in C-HFD Mice, Compared with F-HFD Mice

Gut microbiota analyses revealed characteristic patterns of dysbiosis and biomarker microbial taxa of high abundance in C-HFD mice which showed predictive associations with metabolic pathways and insulin resistance. Next, OGTT and ITT data revealed a significant reduction in glucose tolerance in mice fed with C-HFD compared with mice fed with F-HFD (Figure 5A). Furthermore, the glucose intolerance was reflected in the induction of the area under the curve (AUC) (Figure 5B). In response to insulin during insulin tolerance test (ITT), mice fed with F-HFD showed a greater reduction in glucose levels, compared with C-HFD fed mice, consistent with improved insulin sensitivity. (Figure 5C,D). An increase in the fasting blood glucose levels was observed in the mice fed with C-HFD compared with mice fed with F-HFD (Figure 5E). In addition to fasting hyperglycemia, C-HFD mice, as compared with F-HFD mice, showed a significant reduction in fasting insulin levels (Figure 5F). Higher fasting blood glucose in C-HFD fed mice could be due less prodcution of insulin compared to F-HFD fed mice. Several metabolic hormones including amylin (Figure 5G), gut hormone peptide YY (PYY) (Figure 5H), and gastric inhibitory polypeptide (GIP) (Figure 5I); on the contrary, levels of resistin were upregulated in C-HFD mice compared with F-HFD mice (Figure 5J). However, plasma levels of c-peptide (Figure 5K), glucagon (Figure 5L), leptin (Figure 5M), pancreatic polypeptide (PP) (Figure 5N), ghrelin (Figure 5O), and glucagon-like peptide 1 (GLP-1) (Figure 5P) were relatively low in C-HFD mice compared with F-HFD fed mice but these differences did not reach statistical significance.

## 4. Discussion

Herein, we report for the first time to the best of our knowledge that the intake of sucrose-free C-HFD, but not F-HFD, induces gut microbiome perturbations or dysbiosis that associates with steatohepatitis and insulin resistance in mice. Given that high-sucrose diets are known to induce gut dysbiosis, alteration of the gut epithelial integrity, inflammation, and fatty liver [22,45], using the sucrose-free HFDs in our study was well justified for basically comparing effects of two dietary fats (cocoa butter and fish oil) on the gut microbiome profiles and metabolic derangements in mice. We found that alpha diversity matrices (observed number of OTUs and Fisher’s alpha diversity index) were significantly different between two diets. The C-HFD group had a higher microbial diversity than the F-HFD group. There is substantial evidence to support that different diets shape the gut microbiome differentially, with peculiar effects on host metabolism [46,47]. Our finding that the C-HFD drives a relatively larger gut microbiome diversity, compared with F-HFD, is interesting as the previous studies have tested the microbiome diversity of HFDs against the chow or a standard low-fat diet which contains relatively high fiber content [46,48] and involve some bias as the dietary fiber itself is known to modulate the gut microbiota [49]. Consistent with our results, Wang et al. also found that a high-fat diet increased the alpha diversity of the cecum and colon microbiome in mice, compared with those fed a low-fat control diet [50]. Indeed, high-fat diet is a potential driver of the taxonomical and functional changes that contribute to the biodiversity and dysbiosis of the gut microbiota in mice [51]. PCA, which measures the inter-sample biodiversity (β-diversity) in an unsupervised manner, revealed the distinct clustering between gut microbial communities that we identified between C-HFD and F-HFD fed mice. As expected, the gut microbiota showed a significantly larger interpersonal variation in C-HFD mice, as compared with F-HFD mice, suggesting the distinct functional impacts of two different microbiomes in mice that were fed with two types of high-fat diets, as we later confirmed through clinico-metabolic analyses of these mice.

The C-HFD feeding in our mice promoted a higher relative abundance of *Firmicutes*, *Bacteroidetes*, and *Proteobacteria* while the F-HFD feeding promoted abundances of *Verrucomicrobia*, *TM7*, *Actinobacteria* and *Tenericutes*. Of note, *Firmicutes* and *Bacteroidetes* are the two major phyla that account for almost 90% of the gut microbiota [10] and both these microbial taxa are found to be involved in a wide range of metabolic activities such as carbohydrate metabolism, energy production, amino acid transport, metabolism, and production of short chain fatty acids [52]. We found that the mice in two dietary groups differed regarding the *Firmicutes* to *Bacteroidetes* (F/B) ratios in their gut microbiota. The changes in F/B ratios are considered a marker for the loss of intestinal homeostasis. As opposed to *Bacteroidetes*, the overabundance of Firmicutes in the gut can have deleterious effects. Accordingly, the altered F/B ratio or dysbiosis in the gut has been linked to metabolic impairment and inflammation [53]. The increased F/B ratio is considered a marker of gut dysbiosis and a hallmark of obesity, in both humans [54] and mice [55].

We found that C-HFD feeding in mice induced a higher abundance of Proteobacteria which are normally found only as a minor proportion of gut microbial communities. Proteobacteria may act as opportunistic pathogens and their abnormal expansion is a potentially hazardous signature of the gut dysbiosis linked to several metabolic and inflammatory diseases [56,57]. On the contrary, F-HFD feeding in mice promoted the healthy microbiome, such as *Verrucomicrobia*, *Saccharibacteria (TM7)*, *Actinobacteria*, and *Tenericutes*. *Verrucomicrobia* are mucin-degrading bacteria found in intestinal mucosa that contribute to intestinal health and glucose homeostasis, and play as an interface between the gut microbiome and host tissues [58]. *Saccharibacteria/TM7*, are commensals of healthy oral, gastric, intestinal, and cutaneous microbiome communities [59]. *Saccharibacteria/TM7* play a protective role in inflammatory diseases of mammalian hosts [60]. Notably, Actinobacteria are the core amino acid-metabolizing microbiota in mice with an immunomodulatory potential [61]. The epibiotic association between *Saccharibacteria and Actinobacteria* might be important for the maintenance of heathy gut microbiome or metabolome. *Tenericutes* are known to have anti-inflammatory and immunomodulatory effects by supporting gut tolerance [62]. Interestingly, the C-HFD feeding associated with relative abundances of the OTUs identified with *Bacteroidaceae*, *Lachnospiraceae*, and *Rikenellaceae* families, that are associated with high fat feeding [63,64], and obesity/type 2 diabetes [65]. On the other hand, the F-HFD feeding associated with the OTUs related to *S24-7* or *Muribaculaceae*, *Verrucomicrobiaceae*, *Ruminococcaceae*, and *Porphyromonadaceae*, most of which comprise the major symbiotic gut bacteria in both humans and mice, with key roles in host health by maintaining gut microbiota balance and protection against intestinal inflammation, oxidative stress, and gut epithelial permeability changes [66,67].

While, the F-HFD feeding associated mostly with the abundance of taxa that related to the gut health and homeostasis (*Lactobacillus*, *Coprococcus*, *Anaerotruncus*, *Sporosarcina*, and *Rikenella*) [68], the C-HFD feeding associated with the abundance of *Bacteroides acidifaciens*, *Parabacteroides distasonis*, and *Bacteroides uniformis* and was linked with the suppression of other significant taxa (*Akkermansia muciniphila* and *Mucispirillum schaedleri*), suggesting a plausible pleiotropic effect of cocoa butter on gut microbiome. Next, we found that the highly abundant gut microbial taxa including *Sutterella*, *Anaeroplasma*, *Bacteroides acidifaciens*, *Roseburia*, *Odoribacter*, and *AF12* in C-HFD group were predicted to be associated with various metabolic disorders which is corroborated, at least in part, by previous studies [69,70,71,72,73]. Unlike in F-HFD group, the biomarker gut microbiota in C-HFD group (*Bacteroides uniformis*, *Clostridium*, *Bacteroides acidifaciens*, *Coprococcus*, *Mucispirillum schaedleri*, *Rikenella*, and *Lactobacillus*) had predicted associations with host genes related to metabolic inflammation (*CD44*, *NOD1/2*, *TLR4/5/9*, *NLRP3*, *MC4R*, and *SIRT1*) [74,75,76,77]. These gut microbial taxa are associated with inflammatory signaling including IL-1β, IL-6, IL-8, IL-10, and IL-12, LPS responses, steroids biosynthesis, insulin resistance, and cascades of TLRs, MyD88, NF-κB, NLRP3, ERK1/2, JNK, and MAPK [78,79,80].

A balanced compositional signature of gut microbiota is pivotal to the host health. Gut microbiota are emerging as key players in the regulation of critical metabolic pathways and maintenance of health [81], in humans, mice, and other hosts [82,83,84]. This crosstalk between gut microbiome and host relies mainly on the distinctive biologically active catabolites that are produced and act as metabolic modulators [85] and their alteration may play a role in the impairment of energy/metabolic homeostasis [86]. The ratio between the two dominant bacterial phyla *Firmicutes* and *Bacteroidetes* (F/B ratio) [87] is considered an indicator of the gut microbiota compositional changes and we found the predicted differential patterns of bioactive microbial metabolites and pathways were found between the C-HFD and F-HFD groups and the predicted differential KEGG pathways suggested that microbial community function of gut microbiome was related glucose, lipids, and amino acids metabolism. The changes in gut microbiome and bacterial metabolites impact the expression of host genes related to immune functioning [88,89,90].

These gut microbiome changes were speculated to induce a steatosis-prone inflammatory phenotype in C-HFD mice and as expected, unlike the F-HFD mice, the C-HFD mice showed mixed hepatic steatosis (H&E staining), lobular inflammation and hepatic infiltration of F4/80+ macrophages, and lipid accumulation (Oil red O staining % area) with enhanced liver expression of *Acaca*, *Fasn*, Scd1, Pparγ, *Elovl6*, *Hmgcr*, *Srebp1*, *Ldlr*, *Cpt1a*, *Pparα*, *Tnfα*, and *Ccl2*. *Fatp1* expression was reduced and no significant difference was observed regarding *Cd36*, *Fabp1*, *Abcg1*, and *Mttp* expression in C-HFD and F-HFD mice. To the best of our knowledge, these findings show for the first time a direct link between gut dysbiosis and steatohepatitis in mice that were fed with sucrose-free, C-HFD while the mice fed with F-HFD did not develop those unhealthy metabolic signatures. We speculate that a shift from relatively beneficial to harmful microbiota in the gut leads to an inflammatory and toxic microenvironment that disrupts the gut barrier and exposes the liver to dietary and microbial factors that induce or promote hepatic steatosis and inflammation. Our findings are consistent with other studies deciphering the effects of HFDs on microbiota-associated fatty liver disease in mice [70,91]. In addition to gut microbiome shift, a relative elevation in markers of hepatic fibrosis in C-HFD mice indicates a persistent liver damage in these mice, while no liver damage was observed in F-HFD mice. Our findings of the relationship between gut dysbiosis and liver fibrosis are supported by the study showing that probiotics mixtures administered to maintain the gut microbiome prevented the development of liver fibrosis and endotoxemia in mice fed with high-fat choline-deficient diet [92]. Of note, the gut-liver axis involves a bidirectional communication through the biliary duct, portal vein, and systemic circulation to enable, on one hand, the liver-derived factors to influence the gut microbiota composition and function and, on the other hand, the gut-derived dietary (choline, ethanol, fructose, trimethylamine (TMA)) and microbial factors (LPS, peptidoglycan, bacterial/viral DNA or fungal beta-glucan) or metabolites (short chain fatty acids (SCFA), TMA and its metabolite trimethylamine-*N*-oxide (TMAO), ethanol, acetaldehyde, secondary bile salts deoxycholic acid (DCA), and lithocholic acid (LCA)) to regulate the hepatic glucose and lipid metabolism and bile acid synthesis [93,94]. Moreover, the translocation of microbiota-derived antigens to the liver leads to activation of the innate immunity and liver inflammation [93]. In the gut, endogenous ethanol along with its metabolite acetaldehyde enhances the gut permeability by inducing or upregulating inflammatory cytokines production, reducing AMPs, and disrupting tight junctions. These changes result in gut barrier dysfunction, hepatic translocation of microbiota-derived molecules, induction of inflammatory cytokines, and development of lipogenesis [95].

It is also noteworthy that despite showing abundances of certain bacterial taxa such as Bacteroides acidifaciens, Bacteroides uniformis, and Parabacteroides distasonis which are known for positive effects on glucose metabolism and insulin action [96,97], the C-HFD mice had elevated fasting blood glucose levels and developed glucose intolerance and insulin resistance, which may be due to gut dysbiosis marked by the enhanced F/B ratios. In line with this argument, Foley et al. found that several taxa of *Firmicutes* had a significant positive association with insulin resistance in mice [98]. On the other hand, F-HFD mice which showed abundances of *Verrucomicrobia* and *Actinobacteria* in our study did not develop glucose intolerance and insulin resistance. Interestingly, Foley et al. study also reported no association of *Verrucomicrobia* and *Actinobacteria* with insulin resistance in mice. Besides, the loss of glucose homeostasis in C-HFD mice was accompanied by other critical metabolic derangements including lower circulatory levels of amylin, *PYY*, and *GIP*. Amylin polypeptide hormone is secreted along with insulin by the pancreatic islet β-cells and it regulates blood glucose levels by inhibiting food intake by slowing gastric emptying in rodents [99]. Similarly, *PYY* gut hormone also has the appetite inhibitory action and it reduces food intake to induce postprandial satiety, energy balance and glucose homeostasis in mice [100]. *GIP* is a key player in communicating nutrient intake from the gut to islet, and since *GIPR* is expressed on both α- and β-cells, the GIP-GIPR activity might regulate the optimal levels of both glucagon and insulin to maintain postprandial homeostasis [101]. On the contrary, resistin which is an adipokine known to cause impairment of glucose tolerance and induction of systemic and hepatic insulin resistance was found to be upregulated in C-HFD mice [102]. In addition, relatively lower levels of glucagon, PP, and leptin in C-HFD mice, compared with F-HFD mice, reflect the diverse nature of metabolic impairment that was observed in C-HFD mice.

Nonetheless, our study remains limited by certain caveats such as smaller animal groups, lack of metabolites data from mice, as well as the lack of human data to support generalizability of the results obtained. Therefore, some caution will be needed to interpret these data and further studies will be required to validate our preliminary findings in more diverse settings. Also, since this study used mouse models, it may not be translatable to humans.

## 5. Conclusions

Taken together, our findings suggest that compared with sucrose-free F-HFD feeding, sucrose-free C-HFD feeding in mice leads to the gut microbiome dysbiosis including changes in gut bacterial taxa at levels of relative abundances, composition, and F/B ratio. Importantly, predicted associations between the altered biomarker gut microbiota and the host metabolic pathways were reflected by the pathophysiological impairment in C-HFD mice. The impairment of glucose homeostasis, insulin resistance, and steatohepatitis further imply that contrary to the common notion that cocoa butter may be a healthy nutrient, high intake of this fat, even in the absence of sucrose, could lead to gut dysbiosis and hepatic complications. The below illustration summarizes our key findings (Figure 6).

## Figures and Tables

**Figure 1 nutrients-16-01929-f001:**
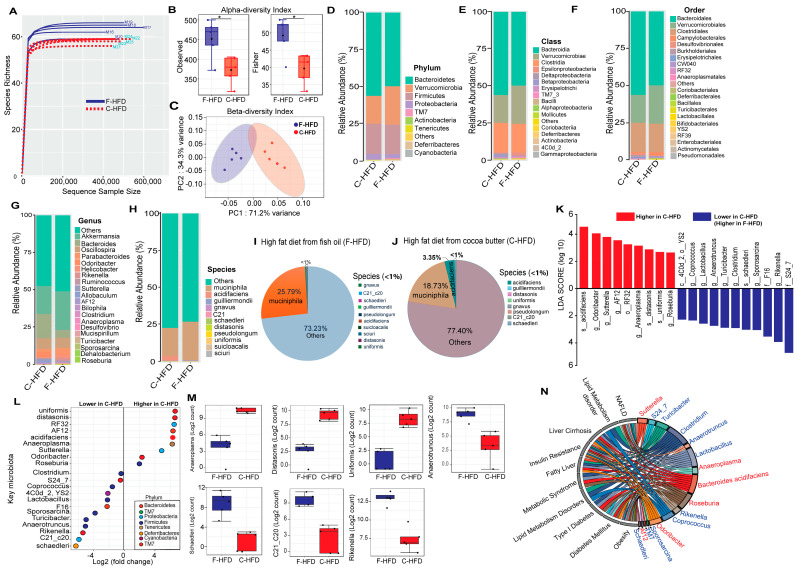
Microbial diversity analysis reveals altered gut microbiota in C-HFD compared with F-HFD-fed mice. (**A**) Influence of 16S rRNA gene sequencing depth on rarefaction analysis was assessed. Rarefaction curves of different samples are shown, with at least 600,000 V3/V4 16S rRNA sequences per sample. Rarefaction curve was created by randomly re-sampling the pool of N samples several times and then plotting the average number of species found in each sample. Typically, it initially grows rapidly as the most common species are found and then slightly flattens out as the rarest species remain to be sampled. Rarefaction curves of different samples collected from C-HFD and F-HFD fed mice are shown (**B**) Alpha diversity across samples from C-HFD and F-HFD fed mice is compared and the differences in alpha diversity metrics of microbial diversity and richness in the gut microbiota of mice fed on two different diets are shown as box plots including the observed number of OTUs and Fisher. Significances marked with an asterisk (*) for *p* ≤ 0.05 refer to the comparisons between C-HFD and F-HFD dietary groups. (**C**) Principal component analysis (PCA) of unweighted UniFrac distances calculated by using 16S rRNA microbiome data set shows that C-HFD data (n = 5, red color) are clustered separately from F-HFD data (n = 5, blue color), *p* value < 0.05. PC1 and PC2 represent the top two principal coordinates that captured most of the variance, with the fraction of variance captured by that coordinate shown as a percentage. Relative abundance of microbial communities associated with C-HFD and F-HFD feeding in mice. (**D**) Phylum (**E**) Class (**F**) Order (**G**) Genus and (**H**) Species level composition of the identified bacterial communities regarding each dietary group. (**I**,**J**) Pie charts of gut microbial species composition in mice fed with C-HFD and F-HFD (n = 5 mice per group). (**K**) Differentially abundant bacteria between C-HFD and F-HFD mice groups are depicted. Linear discriminant analysis (LDA) effect size (LEfSe) comparison of the differentially higher and lower abundance microbial taxa in C-HFD group vs. F-HFD group are shown; 5 mice each group. Log-level changes in LDA score are displayed on the y-axis. Top microbiota of significance by response with an LDA score of >2 as determined using LEfSe. Red bars: taxa found in greater relative abundance in C-HFD and blue bars: taxa found in lower relative abundance in F-HFD (**L**) Dot plot showing the differentially abundant operational taxonomic units (OTUs) where OTUs are grouped by genera along the y-axis and colored as per identified phyla. The x-axis indicates the log2 fold-change in key microbiota (OTUs) in C-HFD-fed mice compared with F-HFD fed mice as the baseline. (**M**) Box plots showing the significant difference concerning fold differential abundance of taxa identified in the gut microbiomes of C-HFD and F-HFD fed mice. (**N**) Predicted interactions between the identified biomarker gut microbiota and the associated metabolic disorders.

**Figure 2 nutrients-16-01929-f002:**
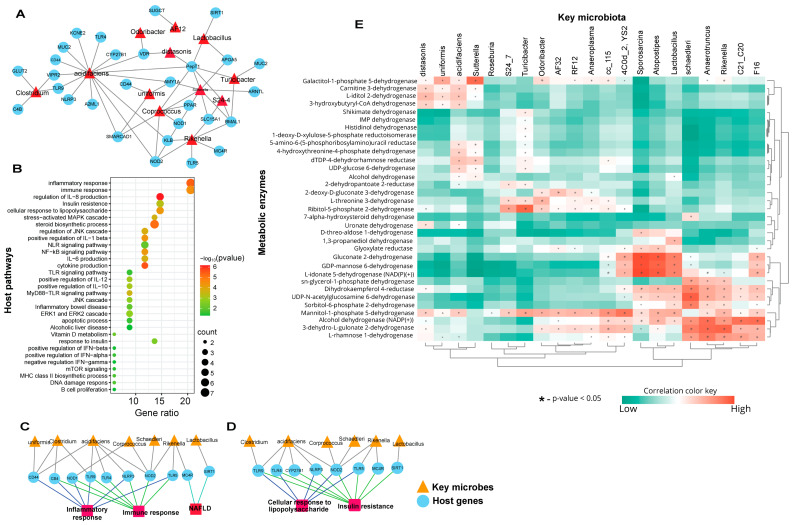
Functional enrichment analysis of the host genes associated with key microbes expressed differentially between C-HFD and F-HFD fed mice. (**A**) Predicted interactions showing associations between the host genes and the differentially expressed biomarker gut microbiota. (**B**) Host pathways enriched for the host genes associated with key gut microbes composition. The size of the dots represents the significance of enrichment for each pathway, and color of the dots denotes the number of host genes in which this pathway is significantly associated with microbiome composition. (**C**,**D**) Networks showing specific gut microbes associated with particular host genes enriched for disease-specific host pathways. Triangular nodes represent the gut microbes, circular nodes represent the host genes, and rectangular nodes represent the pathways. Edge colors represent the host gene-pathway associations. (**E**) Association between microbial genera and metabolic enzymes and metabolites. Correlation between the differentially expressed microbial taxa in C-HFD mice, compared with F-HFD mice, was assessed using linear regression analysis. A heat map of Pearson’s correlations between the differentially expressed gut microbiota and metabolic enzymes or metabolites is shown. Microbial genera are displayed on the x-axis whereas metabolic enzymes and metabolites are shown on the y-axis. Cyan color represents the inverse associations and red color denotes the positive associations. Symbols on the plot represent the significance level with an asterisk (*), denoting the Bonferroni significant associations at *p* < 0.05.

**Figure 3 nutrients-16-01929-f003:**
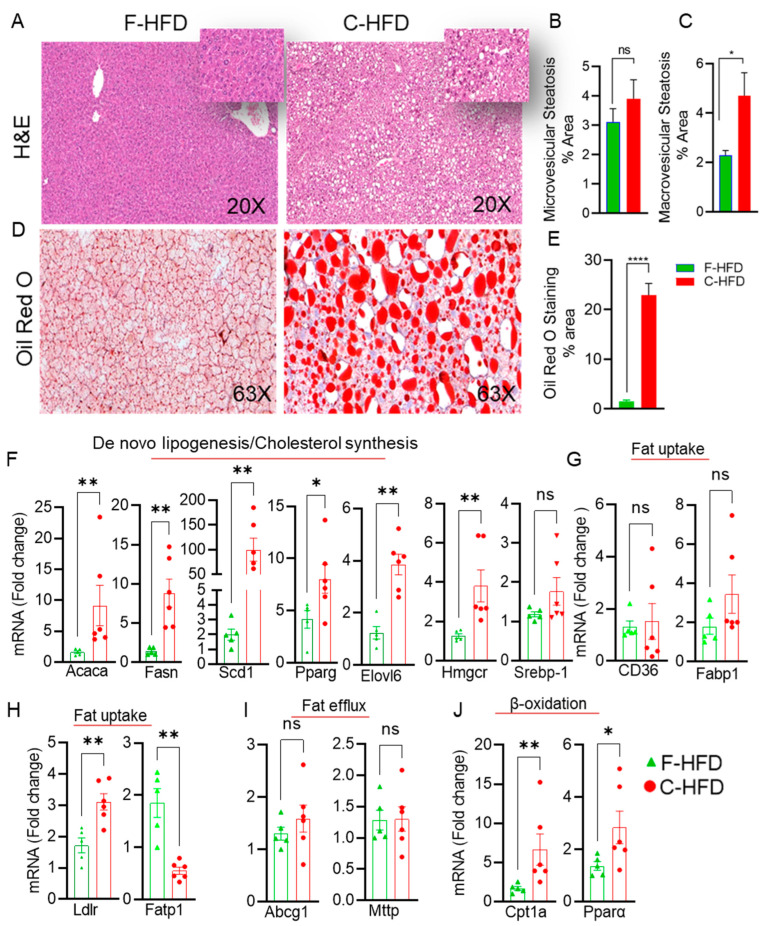
Sucrose-free cocoa butter high-fat diet (C-HFD) feeding in mice, but not fish oil F-HFD, leads to hepatic steatosis and induction of the entire lipogenic program. (**A**) Livers from mice (n = 5–6 per group) fed with C-HFD or F-HFD for 24 weeks were sectioned and stained with H&E (20× magnification). (**B**,**C**) Percentages of microvesicular and macrovesicular stainings are compared in the livers from two dietary mice groups. (**D**) Lipid accumulation as assessed by Oil Red O staining of the liver sections from C-HFD and F-HFD mice is shown. (**E**) % area of Oil Red O staining. (**F**) mRNA expression of genes related to de novo lipogenesis (Acaca, Fasn, Scd1, Elovl6, and Pparg) and cholesterol synthesis (Hmgcr and Srebp1). (**G**,**H**) mRNA expression of genes related to fat uptake (*Cd36*, *Fabp1*, *Ldlr*, and *Fatp1*). (**I**) mRNA expression of genes related to fat efflux (Abcg1 and Mttp). (**J**) mRNA expression of genes related to β-oxidation (*CPT1a* and *Pparα*). Data are presented as mean ± SEM. Significance was determined by using unpaired Student’s *t*-test. * *p* < 0.05 was considered significant; ** *p* < 0.01 was considered highly significant, **** *p* < 0.0001 was considered extremely significant. ns is considered non-significant.

**Figure 4 nutrients-16-01929-f004:**
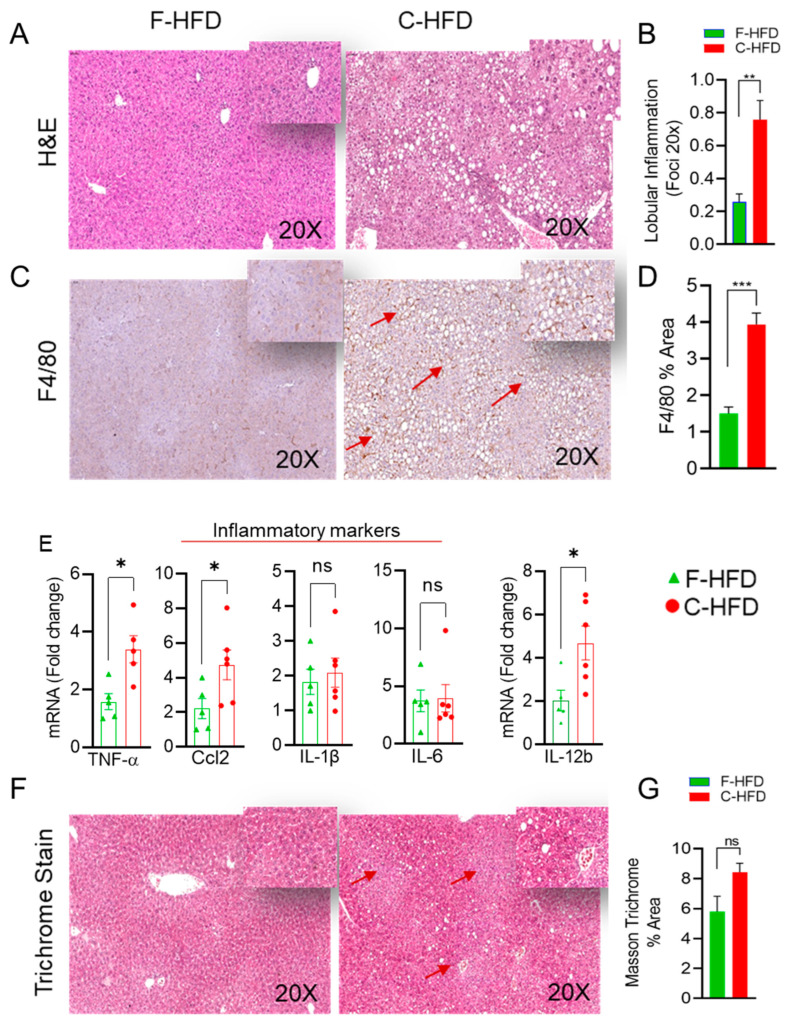
Sucrose-free C-HFD feeding leads to hepatic inflammation. (**A**) Representative images of hematoxylin and eosin (H&E) staining of the liver sections of mice fed with C-HFD and F-HFD for 24 weeks are shown. (**B**) Lobular inflammation is compared between two dietary groups. (**C**) Hepatic infiltrates were stained by immunohistochemistry for macrophage (F4/80) markers. (**D**) Macrophage infiltration (F4/80-positive % area) is compared between two dietary groups of mice. (**E**) Comparative mRNA expression of inflammatory marker genes (Tnfa, Ccl2, Il1b, Il6, and Il12b), as determined by quantitative RT-qPCR, in C-HFD and F-HFD mice groups. (**F**,**G**) Comparison of collagen deposition as assessed by Masson’s trichrome staining in the liver sections of C-HFD and F-HFD mice. Data are presented as mean ± SEM values. Significance was determined by using unpaired Student’s *t*-test. * *p* < 0.05, ** *p* < 0.01, *** *p* < 0.001 were considered significant. ns was considered non-significant.

**Figure 5 nutrients-16-01929-f005:**
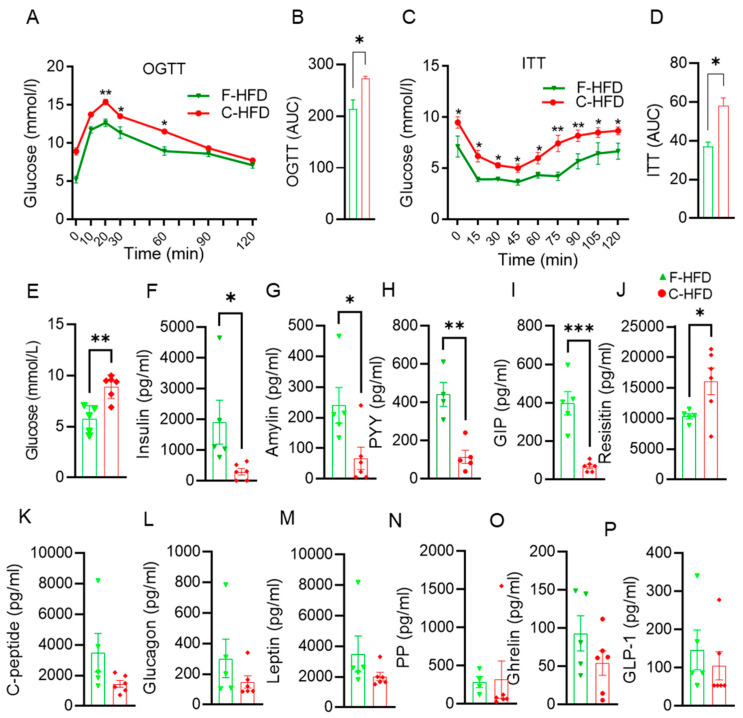
C-HFD feeding impairs glucose and insulin tolerance as well as C-HFD ameliorates glucose tolerance and insulin resistance in mice. (**A**,**B**) Oral glucose tolerance test (OGTT) at 21 weeks of C-HFD and F-HFD feeding and the area under curve (AUC) are compared for C-HFD and F-HFD mice. (**C**,**D**) Insulin tolerance test (ITT, i.p.) at 22 weeks of C-HFD and F-HFD feeding and the area under curve (AUC) are compared for C-HFD and F-HFD mice. (**E**,**F**) Fasting plasma glucose and insulin levels in mice at 24 weeks of C-HFD and F-HFD feeding. (**G**–**P**) Fasting blood metabolic markers profiles were determined using Milliplex assay kits and levels of c-peptide, glucagon, amylin, leptin, PYY, PP, ghrelin, GLP-1, GIP, and resistin are compared between C-HFD and F-HFD mice. Data are presented as mean ± SEM values. * *p* < 0.05 was considered significant; ** *p* < 0.01 was considered highly significant, *** *p* < 0.001 was considered extremely significant.

**Figure 6 nutrients-16-01929-f006:**
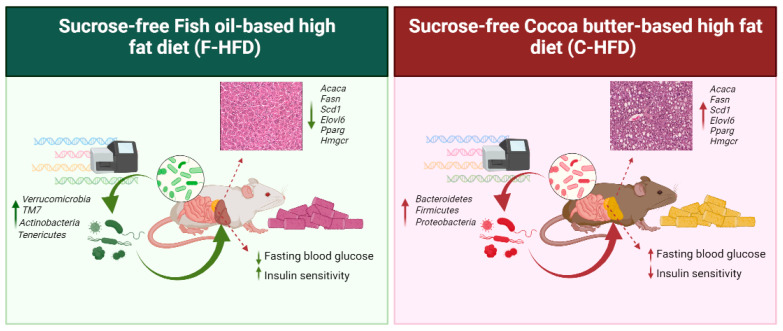
Graphical illustration presents the summary of our key findings.

## Data Availability

The data that support the findings of this study are available on request from the corresponding author. The dataset analysed during the current study is available in the NCBI repository, under the BioProject ID PRJNA1007291 (at https://dataview.ncbi.nlm.nih.gov/object/PRJNA1007291?reviewer=47jqvq8ahucdptd0m8nid56v9u (accessed on 1 April 2023)). The accession number for each sample is provided in Appendix A.

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
