# Peer review of "Gut Dysbiosis Shaped by Cocoa Butter-Based Sucrose-Free HFD Leads to Steatohepatitis, and Insulin Resistance in Mice"

_nutrients, 2024, doi:10.3390/nu16121929_

Round 1

Reviewer 1 Report

Comments and Suggestions for Authors

In the manuscript entitled “Gut dysbiosis shaped by Cocoa butter-based sucrose-free HFD leads to steatohepatitis, and insulin resistance in mice”, the authors investigated the effects of a sucrose-free cocoa butter-based high-fat diet (C-HFD) on health parameters of mice. The authors showed that the gut microbiota exhibited significant variation between C-HFD and sucrose-free fish oil-based high-fat diet (F-HFD) mice, indicating distinct functioanl capabilities of the two microbiomes. To further explore the effects of altered gut microbiome caused by C-HFD, the researchers analyzed functional predictions, which was associated with reduced immunity to pathogen. Furthermore, the authors found that the steatosis-prone inflammatory phenotype and increased insulin resistance were more pronounced in C-HFD mice compared to F-HFD mice. Through these regults, the authors suggest that altered gut microbiota composition due to C-HFD contributes to disease. This study reveals the possibility that C-HFD induces gut dysbiosis, leading to glucose homeostasis, insulin resistance, and steatohepatitis. Following are my concerns.

Major comments

1. I believe it is better to simplify the abstract and discussion by retaining only essential parts, such as the research aim, the association between diet and phenotype and microbiome. It would help authors focus on providing essential information to facilitate understanding of the following results. Additionally, in the conclusion section of the abstract, the statement that the gut microbiome leads to insulin is incorrect. Please rewrite this statement more accurately.

2. Many key citations are missing and they need to add multiple citations including the following ones. Add citations to the describing the effect of gut microbiome on host in a variety of species, including mice and humans (Lee et al., PMID: 37853687, Thompson et al., PMID: 37494472, Fan et al., PMID: 37794596). In addition, add citation in the sentence describing the role of host genes associated gut microbiota on immune system (Piao et al., PMID: 37587649, Gabbia et al., PMID: 38132297, Lee et al., PMID: 34887364).

3. Several figures should be integrated and modified. For example, I recommend that the authors integrate figure 1 to 3 into one figure showing how C-HFD and F-HFD alter the composition of the microbiome. In addition, I recommend that the authors integrate figure 4 and 5 showing how the altered gut microbiome correlates with the health of the host. Furthermore, the authors need to describe the significance and meaning of data in the Result section to highlight the finding of this study. Again this will make the paper thin and another example why the paper looks incomplete.

4. I think this paper has interesting phenomenon but mechanistically it looks like an incomplete paper unfortunately. The authors suggest that sucrose free C-HFD feeding in mice leads to dysbiosis, hepatic steatosis, inflammation, and insulin resistance. However, there seems to be a lack of evidence regarding the correlation between the altered gut microbiome caused by C-HFD feeding and these phenomena. To further support the authors’ statement, I think they should investigate which factors mediate these phenomena or if there are any previous data to directly prove this correlation. If the authors have references for this correlation, please include the citation.

Minor comments

1. Please be consistent with the format of graph in all figures, including supplementary (font, x- and y- axis, symble shapes etc).

2. In figure 1 to 8, such as H&E and Oil Red O staining, the authors need to incorporate the quantification and statistic method.

3. I wonder how many iterations of the experiment the authors conducted. If the authors conducted the experiment with only one set of mice, the significance of the results seems to be lacking.

Comments on the Quality of English Language

Mostly fine.

Author Response

Response to Reviewer 1 Comments

Ref: Manuscript ID nutrients-2989252

Title: Gut dysbiosis shaped by Cocoa butter-based sucrose-free HFD leads to steatohepatitis, and insulin resistance in mice

We thank the reviewer for thoughtful comments. Please see below the point-by-point responses to the comments or concerns raised.

Comments and Suggestions for Authors

In the manuscript entitled “Gut dysbiosis shaped by Cocoa butter-based sucrose-free HFD leads to steatohepatitis, and insulin resistance in mice”, the authors investigated the effects of a sucrose-free cocoa butter-based high-fat diet (C-HFD) on health parameters of mice. The authors showed that the gut microbiota exhibited significant variation between C-HFD and sucrose-free fish oil-based high-fat diet (F-HFD) mice, indicating distinct functioanl capabilities of the two microbiomes. To further explore the effects of altered gut microbiome caused by C-HFD, the researchers analyzed functional predictions, which was associated with reduced immunity to pathogen. Furthermore, the authors found that the steatosis-prone inflammatory phenotype and increased insulin resistance were more pronounced in C-HFD mice compared to F-HFD mice. Through these regults, the authors suggest that altered gut microbiota composition due to C-HFD contributes to disease. This study reveals the possibility that C-HFD induces gut dysbiosis, leading to glucose homeostasis, insulin resistance, and steatohepatitis. Following are my concerns.

Major comments

  1. I believe it is better to simplify the abstract and discussion by retaining only essential parts, such as the research aim, the association between diet and phenotype and microbiome. It would help authors focus on providing essential information to facilitate understanding of the following results. Additionally, in the conclusion section of the abstract, the statement that the gut microbiome leads to insulin is incorrect. Please rewrite this statement more accurately.

Authors’ response: Thanks to the reviewer for guiding comments. Accordingly, the abstract and discussion sections are now modified in light of kind suggestions. The abstract conclusion is also re-phrased for necessary correction as required.

  1. Many key citations are missing and they need to add multiple citations including the following ones. Add citations to the describing the effect of gut microbiome on host in a variety of species, including mice and humans (Lee et al., PMID: 37853687, Thompson et al., PMID: 37494472, Fan et al., PMID: 37794596). In addition, add citation in the sentence describing the role of host genes associated gut microbiota on immune system (Piao et al., PMID: 37587649, Gabbia et al., PMID: 38132297, Lee et al., PMID: 34887364).

Authors’ response: Thanks for the suggestions. The recommended references have now been added to the relevant parts in the discussion.

  1. Several figures should be integrated and modified. For example, I recommend that the authors integrate figure 1 to 3 into one figure showing how C-HFD and F-HFD alter the composition of the microbiome. In addition, I recommend that the authors integrate figure 4 and 5 showing how the altered gut microbiome correlates with the health of the host. Furthermore, the authors need to describe the significance and meaning of data in the Result section to highlight the finding of this study. Again this will make the paper thin and another example why the paper looks incomplete.

Authors’ response: Figures 1-3 have been integrated into Figure 1 as suggested. Figures 4 and 5 were integrated and new figure number is 2. As suggested, the significance and meaning of the data are also added to the Results section.

  1. I think this paper has interesting phenomenon but mechanistically it looks like an incomplete paper unfortunately. The authors suggest that sucrose free C-HFD feeding in mice leads to dysbiosis, hepatic steatosis, inflammation, and insulin resistance. However, there seems to be a lack of evidence regarding the correlation between the altered gut microbiome caused by C-HFD feeding and these phenomena. To further support the authors’ statement, I think they should investigate which factors mediate these phenomena or if there are any previous data to directly prove this correlation. If the authors have references for this correlation, please include the citation.

Authors’ response: We agree with the reviewer that there remains a lack of direct evidence of the correlation between altered gut microbiome and the pathotypes of steatosis, inflammation and insulin resistance in the mice fed with C-HFD. In addressing this aspect, relevant references have been added to the revised manuscript.

Minor comments

  1. Please be consistent with the format of graph in all figures, including supplementary (font, x- and y- axis, symble shapes etc).

Authors’ response: Done

  1. In figure 1 to 8, such as H&E and Oil Red O staining, the authors need to incorporate the quantification and statistic method.

Authors’ response: Done-(Supplementary file)

  1. I wonder how many iterations of the experiment the authors conducted. If the authors conducted the experiment with only one set of mice, the significance of the results seems to be lacking.

Authors’ response: 5-6 mice in each set of experiment, and the data are representative of 2 independent sets of experiments.

Comments on the Quality of English Language

Mostly fine.

Reviewer 2 Report

Comments and Suggestions for Authors

The authors studied the effects of cocoa butter on liver in mice. It is very detailed and informative, especially on intestinal microbiome.

I would like to make a few corrections.

LL.81-88, Lard and palm oil are high in saturated fatty acids and bad for your health. Fish oil and olive oil are high in unsaturated fatty acids and good for your health. Then, on line 85, it suddenly says, "Cocoa butter and fish oil are good for your health. First of all, it needs to be clearly stated whether cocoa butter is high in saturated fatty acids or unsaturated fatty acids. And it is not correct for the purpose of the paper to say that cocoa butter is as healthy as fish oil.

Lines 506-519, the figure numbers do not match the description.

What were the changes in insulin levels during the OGTT? If insulin is always low when it is given in ITT, it is thought to lower blood glucose levels. Insulin resistance is when insulin is being secreted but blood glucose levels do not decrease.

Author Response

Response to Reviewer 2 Comments

Ref: Manuscript ID nutrients-2989252

Title: Gut dysbiosis shaped by Cocoa butter-based sucrose-free HFD leads to steatohepatitis, and insulin resistance in mice

We thank the reviewer for thoughtful comments. Please see below the point-by-point responses to the comments or concerns raised.

Comments and Suggestions for Authors

The authors studied the effects of cocoa butter on liver in mice. It is very detailed and informative, especially on intestinal microbiome.

I would like to make a few corrections.

LL.81-88, Lard and palm oil are high in saturated fatty acids and bad for your health. Fish oil and olive oil are high in unsaturated fatty acids and good for your health. Then, on line 85, it suddenly says, "Cocoa butter and fish oil are good for your health. First of all, it needs to be clearly stated whether cocoa butter is high in saturated fatty acids or unsaturated fatty acids. And it is not correct for the purpose of the paper to say that cocoa butter is as healthy as fish oil.

Authors Response: We thank the reviewer for technical comments and necessary corrections have been made in the revised manuscript as required.

Lines 506-519, the figure numbers do not match the description.

Authors Response: Corrected.

What were the changes in insulin levels during the OGTT? If insulin is always low when it is given in ITT, it is thought to lower blood glucose levels. Insulin resistance is when insulin is being secreted but blood glucose levels do not decrease.

Authors Response: In insulin sensitive animals, during OGTT, there was low fasting insulin and the insulin peak was sharp and early, indicating effective insulin action and an efficient glucose uptake by tissues. On the contrary, in insulin-resistant animals, there was higher fasting insulin and the insulin peak was higher and delayed, indicating that more insulin was required over a longer time to achieve glucose homeostasis. Also, in insulin-sensitive animals, there was a rapid return to baseline insulin levels while there were prolonged and elevated post-glucose exposure insulin levels observed in insulin-resistant animals.

Reviewer 3 Report

Comments and Suggestions for Authors

In their manuscript, the authors analyze an extremely important issue: the impact of a high-fat diet, here the impact of two different fats  (cocoa butter and fish oil), components of a healthy diet, on intestinal bacteria and their dysbiosis as well as steatohepatitis and insulin resistance in mice. The manuscript has proper composition and is well written. The material and methods were presented in a clear and appropriate manner, also the results were presented in a logical and comprehensive manner. Below are some minor comments:

1. please expand the abbreviations in the first place they appear

2.  move fig.2 above its description - I suggest enlarging the image slightly to improve its readability

3. completed the decoding for "**/***" in Fig.6 and 7 as in Fig.8

Author Response

Response to Reviewer 3 Comments

Ref: Manuscript ID nutrients-2989252

Title: Gut dysbiosis shaped by Cocoa butter-based sucrose-free HFD leads to steatohepatitis, and insulin resistance in mice

We thank the reviewer for thoughtful comments. Please see below the point-by-point responses to the comments or concerns raised.

Comments and Suggestions for Authors

In their manuscript, the authors analyze an extremely important issue: the impact of a high-fat diet, here the impact of two different fats  (cocoa butter and fish oil), components of a healthy diet, on intestinal bacteria and their dysbiosis as well as steatohepatitis and insulin resistance in mice. The manuscript has proper composition and is well written. The material and methods were presented in a clear and appropriate manner, also the results were presented in a logical and comprehensive manner. Below are some minor comments:

  1. please expand the abbreviations in the first place they appear

Authors’ Response: Thanks for your kind suggestion. All abbreviations are now described in the first place as suggested.

  1. move fig.2 above its description - I suggest enlarging the image slightly to improve its readability

Authors’ Response: Done.

  1. completed the decoding for "**/***" in Fig.6 and 7 as in Fig.8

Authors’ Response:  Done as suggested.

Round 2

Reviewer 2 Report

Comments and Suggestions for Authors

Thank you for revising your manuscript.

I have one question.

Your answer is;

In insulin sensitive animals, during OGTT, there was low fasting insulin and the insulin peak was sharp and early, indicating effective insulin action and an efficient glucose uptake by tissues. On the contrary, in insulin-resistant animals, there was higher fasting insulin and the insulin peak was higher and delayed, indicating that more insulin was required over a longer time to achieve glucose homeostasis. Also, in insulin-sensitive animals, there was a rapid return to baseline insulin levels while there were prolonged and elevated post-glucose exposure insulin levels observed in insulin-resistant animals.

If so, why in Fig.5F, blood insulin concentration is lower in C-HFD mice?

Author Response

Response to Reviewer 2 Comments

Ref: Manuscript ID nutrients-2989252

Title: Gut dysbiosis shaped by Cocoa butter-based sucrose-free HFD leads to steatohepatitis, and insulin resistance in mice

We thank the reviewer for thoughtful comments. Please see below the point-by-point responses to the comments or concerns raised.

Comments and Suggestions for Authors

omments and Suggestions for Authors

Thank you for revising your manuscript.

I have one question.

Your answer is;

In insulin sensitive animals, during OGTT, there was low fasting insulin and the insulin peak was sharp and early, indicating effective insulin action and an efficient glucose uptake by tissues. On the contrary, in insulin-resistant animals, there was higher fasting insulin and the insulin peak was higher and delayed, indicating that more insulin was required over a longer time to achieve glucose homeostasis. Also, in insulin-sensitive animals, there was a rapid return to baseline insulin levels while there were prolonged and elevated post-glucose exposure insulin levels observed in insulin-resistant animals.

If so, why in Fig.5F, blood insulin concentration is lower in C-HFD mice?

Authors’ response:

I apologize for the confusion. We improved the subtitle as:  3.4. Impaired glucose tolerance in C-HFD mice, compared with F-HFD mice. We revised the results lines related to Figure 5F as :In response to insulin during insulin tolerance test (ITT), mice fed with F-HFD showed a greater reduction in glucose levels, compared with C-HFD fed mice, consistent with improved insulin sensitivity.   (Fig. 5C-D). An increase in the fasting blood glucose levels was observed in the mice fed with C-HFD compared with mice fed with F-HFD (Fig.5E). In addition to fasting hyperglycemia, C-HFD mice, as compared with F-HFD mice, showed a significant reduction in fasting insulin levels (Fig.5F). Higher fasting blood glucose in C-HFD fed mice could be due less production of insulin compared to F-HFD fed mice.
